# Discourse Structures Guided Fine-grained Propaganda Identification

**Yuanyuan Lei and Ruihong Huang**
Department of Computer Science and Engineering
Texas A&M University, College Station, TX
{yuanyuan, huangrh}@tamu.edu

## Abstract

Propaganda is a form of deceptive narratives that instigate or mislead the public, usually with a political purpose. In this paper, we aim to identify propaganda in political news at two fine-grained levels: sentence-level and token-level. We observe that propaganda content is more likely to be embedded in sentences that attribute causality or assert contrast to nearby sentences, as well as seen in opinionated evaluation, speculation and discussions of future expectation. Hence, we propose to incorporate both local and global discourse structures for propaganda discovery and construct two teacher models for identifying PDTB-style discourse relations between nearby sentences and common discourse roles of sentences in a news article respectively. We further devise two methods to incorporate the two types of discourse structures for propaganda identification by either using teacher predicted probabilities as additional features or soliciting guidance in a knowledge distillation framework. Experiments on the benchmark dataset demonstrate that leveraging guidance from discourse structures can significantly improve both precision and recall of propaganda content identification.[1]

## 1 Introduction

Propaganda refers to a type of misleading and deceptive information used to promote or publicize a certain political point of view (Lasswell, 1927; Henderson, 1943; Stanley, 2015; Rashkin et al., 2017). This information is often manipulated in a strategic way to shape societal beliefs (Rashkin et al., 2017; Barron-Cedeno et al., 2019). Propaganda can be harmful to both individuals and society as a whole, such as disseminating false information, inciting people's perceptions, leading to conflicts, perpetuating prejudices, impeding democracy process etc. (Bernays, 2005; Stanley, 2015; Little,

---

[1]The code and data link: https://github.com/yuanyuanlei-nlp/propaganda_emnlp_2023

2017). Despite its misleading nature and harmful effects, propaganda can be pervasive in political news, and has the potential to reach very large audiences (Glowacki et al., 2018; Tardaguila et al., 2018). Hence, developing intelligent models to identify propaganda in political news is important and necessary.

Instead of detecting propaganda at the level of articles (Horne et al., 2018; De Sarkar et al., 2018; Rashkin et al., 2017; Rubin et al., 2016), this paper focuses on identifying propaganda at fine-grained levels: sentence-level and token-level. Through extracting the sentence or fragment imbued with propaganda, our aim is to accurately locate the propagandistic content and thereby provide detailed interpretable explanations. Propaganda content not only presents unverified or even false information, but also employs a variety of argumentation strategies to convince the readers (Yu et al., 2021). Therefore, identifying propaganda at fine-grained levels still remains a difficult and challenging task, and requires profound understanding of broader context in an article (Da San Martino et al., 2019b).

First, we observe that propaganda can be injected into sentences that attribute causality or assert contrast to nearby sentences. Take the article in Figure 1 as an example, the second sentence (S2) makes an illogical and misleading deduction from its preceding sentence: *This suggested giving advice on how to prevent Jihadist attacks is now against community standards.* Propaganda content such as S2 usually disseminate the misinformation by leveraging causal relations, either by inferring baseless reasons or deducting speculative consequences. In addition, propaganda content can also utilize contrast relation to raise doubt or challenge credibility. For example, the last sentence (S4) casts doubts towards its previous sentence: *why would Facebook remove such a page and never explain why?* Through the strategy of contrasting, the author aims to undermine the credibility of Facebook and

| Example Article 1: sentences demonstrating locally **causal** or **comparison** discourse relations are more likely to carry propaganda | | |
|---|---|---|
| index | sentences | discourse relation |
| S1 | On the 17th Anniversary of the 9/11, Facebook informed Jamie that he was suspended for 30 days due to his article -- 9 Steps to Successfully Counter Jihad, violated their "community standards." | |
| S2 | This suggested, of course, that giving advice on how to prevent another 9/11, and all other Jihadist attacks against America, is now against Facebook's 'community standards'. | Causal |
| S3 | The Counter Jihad Coalition's (CJC) Facebook page, a human rights group that is dedicated to protecting America from Sharia oppression, which Jamie helped run, was removed years ago with absolutely no explanation. | |
| S4 | The question remains: why would Facebook remove such a page, let alone in such a fascistic manner -- and never explain why? | Comparison |

Figure 1: An example article containing propaganda sentences. Propaganda sentences are highlighted in red. S2 is a propaganda sentence showing deduction. S4 is a propaganda sentence proposing challenge or doubt. Their discourse relations with nearby sentence are shown in right column.

| Example Article 2: propaganda content is more likely to be embedded into **opinionated evaluations** and **expectation speculations** | | |
|---|---|---|
| index | sentences | news discourse role |
| S1 | Immigrations and Customs Enforcement (ICE) arrested more than 150 illegals in the San Francisco Bay Area this week | Main Event |
| S2 | It was reported that about half of snared fugitives with convictions for assault and battery, crimes against children, and weapons charges. | Current Context |
| S3 | The Democratic mayor of Oakland, Libby Schaaf, tweeted that ICE would be conducting the raids in the Bay Area. | Current Context |
| S4 | ICE Deputy Director Thomas Homan said in a statement Tuesday, "The Oakland mayor's decision to publicize her suspicions about ICE operations increased risk for my officers and alerted criminal aliens — making clear that this reckless decision was based on her political agenda" | Evaluation |
| S5 | "Unlike the politicians who attempt to undermine ICE's critical mission, our officers will continue to fulfill their sworn duty to protect public safety." | Expectation |

Figure 2: Another example article containing propaganda sentences, and the corresponding news discourse role of each sentence. Propaganda sentences are highlighted in red. Propaganda content is more likely to be embedded into opinionated evaluations and expectation speculations.

thereby incite public protest. Accordingly, we propose that understanding the discourse relations of a sentence with its nearby sentences in the local context can enable discovery of propaganda contents.

Moreover, we observe that propaganda content or deceptive narratives is more likely to be embedded into opinionated evaluations or expectation speculations. In contrast, sentences describing factual occurrences are less likely to carry propaganda. Take the article in Figure 2 as an example, the first three sentences, either reporting the main event or describing the current context triggered by the main event, all provide the readers with factual occurrences devoid of deceptive content. However, in the succeeding sentence (S4), the author includes a quotation to express emotional assessments: *this reckless decision was based on her political agenda*. Propaganda sentences such as S4 always convince the readers and influence their mindset by inserting opinionated evaluations. Furthermore, the author speculates future expectations in the next sentence (S5) that *ICE officers will continue to protect public safety*. Propaganda sentences such as S5 usually promise a bright yet unprovable future with the aim of gaining trust and support. Therefore, we propose that understanding the discourse role of a sentence in telling a news story can help reveal propaganda.

Motivated by the above observations, we propose to incorporate both local and global discourse structures for propaganda identification. Specifically, we establish two teacher models to recognize PDTB-style discourse relations between a sentence and its nearby sentences (Prasad et al., 2008), as well as identify one of eight common news discourse roles for each sentence based upon news discourse structure (Choubey et al., 2020). We further devise two approaches to effectively incorporate the two types of discourse structures for propaganda identification. The first approach concatenates the predicted probabilities from two teacher models as additional features. The second approach develops a more sophisticated knowledge distillation framework, where we design a response-based distillation loss to mimic the prediction behavior of teacher models, as well as a feature relation-based distillation loss to seek guidance from the embeddings generated by teacher models. The response-based and feature relation-based distillation mutually complement each other, acquiring an enhanced guidance from discourse structures. Experiments on the benchmark dataset demonstrate the effectiveness of our approaches for leveraging discourse structures, with both precision and recall improved. The ablation study validates the necessity and syn-

| | Comparison | Contingency | Temporal | Expansion | Total |
|---|---|---|---|---|---|
| propaganda | **102 (35.66)** | **146 (40.56)** | 18 (18.18) | 337 (32.13) | 620 (30.48) |
| benign | 184 (64.34) | 214 (59.44) | 81 (81.82) | 712 (67.87) | 1414 (69.52) |

Table 1: The number (ratio) of propaganda and benign sentences that have each of the four discourse relations with nearby sentences. The ratio values higher than the overall ratio in the rightmost column are shown in **bold**.

ergy between local and global discourse structures.

## 2 Discourse Structures

In this section, we explain the details for the two discourse structures: discourse relation based on PDTB-style relations, and discourse role that draws upon news discourse structure. We also perform a statistical analysis to verify our empirical observations, and introduce the procedure of constructing teacher models for both discourse structures.

### 2.1 Discourse Relations

#### 2.1.1 PDTB Discourse Structure

The Penn Discourse Treebank (PDTB) discourse structure (Prasad et al., 2008) interprets the discourse relation between adjacent sentences in news articles into four types: 1). Comparison highlights prominent differences between two arguments, and represents the relation of contrasting or concession. 2). Contingency indicates two arguments causally influence each other, and represents a cause-and-effect or conditional relationship. 3). Temporal captures the temporal or chronological relationship between two arguments, such as precedence, succession, or simultaneously. 4). Expansion covers relations of elaborating additional details, providing explanations, or restating narratives.

#### 2.1.2 Statistical Analysis

To validate the correlation between propaganda and discourse relations, we also conduct a statistical analysis on the validation set of propaganda dataset (Da San Martino et al., 2019b), where we run the model of classifying discourse relations. Table 1 shows the ratio of propaganda sentences that have each of the four discourse relations with nearby sentences. The numerical analysis confirms our observation: sentences that exhibit contingency and comparison relations with adjacent sentences are more prone to containing propaganda, whereas sentences that narrate events in a chronological order significantly contain less propaganda.

#### 2.1.3 Teacher Model for Discourse Relation

We train the teacher model for discourse relations by using Longformer (Beltagy et al., 2020) as the basic language model. The sentence pair embedding is the concatenation of hidden states at the two sentences start tokens . A two-layer neural network is built on top of the pair embedding to predict discourse relations into comparison, contingency, temporal, or expansion. The model is trained on PDTB 2.0 data (Prasad et al., 2008) that annotates both explicit and implicit relations between adjacent sentences. Considering propaganda sentences can be connected with nearby sentences with or without discourse connectives explicitly shown, we utilize both explicit and implicit discourse relations data for training.

Given a pair of sentences from the propaganda article, the local discourse relation teacher model generates the predicted probability of four relations between $i$-th sentence and its nearby sentence as:

$$P_i^{local} = (P_{i1}^{local}, P_{i2}^{local}, P_{i3}^{local}, P_{i4}^{local}) \quad (1)$$

### 2.2 Discourse Role

#### 2.2.1 News Discourse Structure

The news discourse structure (Choubey et al., 2020) categorizes the discourse role of each sentence in news article into three broad types and eight subtypes: 1). main event contents contain two subtypes, Main event (M1) and Consequence (M2), and cover sentences that describe the main event and their immediate consequences which are often found inseparable from main events. 2). context-informing contents have two subtypes, Previous Event (C1) and Current Context (C2), and cover sentences that explain the context of the main event, including recent events and general circumstances. 3). additional supportive contents have four subtypes, describing past events that precede the main event in months and years (Historical Event (D1)) or unverifiable fictional situations (Anecdotal Event (D2)), or opinionated contents including reactions from immediate participants, experts, known personalities as well as journalists or news sources (Evaluation (D3)), except speculations and projected consequences referred as Expectation (D4).

| | M1 | M2 | C1 | C2 | D1 | D2 | D3 | D4 | Total |
|---|---|---|---|---|---|---|---|---|---|
| propaganda | 66 (28.08) | 0 (none) | 9 (19.56) | 71 (17.07) | **51 (33.12)** | 3 (12.00) | **335 (42.84)** | **55 (36.18)** | 620 (30.48) |
| benign | 169 (71.92) | 0 (none) | 37 (80.44) | 345 (82.93) | 103 (66.88) | 22 (88.00) | 447 (57.16) | 97 (63.82) | 1414 (69.52) |

Table 2: The number (ratio) of propaganda and benign sentences under each of the eight news discourse role types. The rightmost column shows the overall number (ratio). The ratio values higher than the overall ratio are shown in **bold**. M1: Main Event, M2: Consequence, C1: Previous Context, C2: Current Context, D1: Historical Event, D2: Anecdotal Event, D3: Evaluation, D4: Expectation

### 2.2.2 Statistical Analysis

To verify the correlation between propaganda and news discourse structure, we perform a statistical analysis on the validation set of propaganda dataset (Da San Martino et al., 2019b), where we run the model of profiling news discourse structure (Choubey and Huang, 2021). Table 2 presents the ratio of propaganda sentences across the eight news discourse roles. The numerics validate our observations: propaganda is more likely to be embedded into sentences expressing opinions or evaluations (D3), speculating future expectations (D4), or fabricating historical background (D1). Conversely, sentences describing factual occurrences, such as reporting main event (M1) or informing context (C1, C2) are less likely to carry propaganda.

### 2.2.3 Teacher Model for Discourse Role

We follow the same framework in the current state-of-art model of profiling news discourse structure (Choubey and Huang, 2021), where an actor-critic model is developed that selects between the standard REINFORCE (Williams, 1992) algorithm or imitation learning for training actor. Additionally, we replace the ELMo word embeddings (Peters et al., 2018) with Longformer language model (Beltagy et al., 2020), which generates contextualized embeddings for long documents based on transformer (Vaswani et al., 2017) and provides further improvements to the current state-of-the-art.

Given a candidate propaganda article consisting of $n$ sentences, the global discourse role teacher model generates the predicted probability of eight discourse roles for $i$-th sentence as:

$$P_i^{global} = (P_{i1}^{global}, P_{i2}^{global}, ..., P_{i8}^{global}) \quad (2)$$

## 3 Fine-grained Propaganda Identification

In order to incorporate the two types of discourse structures into propaganda identification, we further devise two methods: a feature concatenation model and a knowledge distillation model. Figure 3 illustrates the framework of knowledge distillation.

Considering the news articles are typically long, we utilized Longformer (Beltagy et al., 2020) as the basic language model to encode the entire article. Given a candidate propaganda article consisting of $n$ sentences, sentence embeddings $(s_1, s_2, ..., s_n)$ are initialized as the hidden state at sentence start tokens . The $i$-th sentence contains $m$ tokens, and its tokens embeddings are $(w_{i1}, w_{i2}, ...w_{im})$.

### 3.1 Feature Concatenation Model

The feature concatenation model directly concatenates the predicted probabilities generated by the two teacher models as additional features, since they contain the discourse structures information. The updated feature vectors for $i$-th sentence and its $j$-th token in the two fine-grained tasks are:

$$\hat{s}_i = s_i \oplus P_i^{local} \oplus P_i^{global}$$
$$\hat{w}_{ij} = w_{ij} \oplus P_i^{local} \oplus P_i^{global} \quad (3)$$

where $\oplus$ denotes feature concatenation, $P_i^{local}$ and $P_i^{global}$ are probabilities of discourse relations and discourse roles predicted by two teacher models.

Additionally, a two-layer classification head is built on top of the updated embedding to make prediction. The cross-entropy loss is used for training.

### 3.2 Knowledge Distillation Model

The knowledge distillation model constructs additional learning layers to learn local discourse relation and global discourse role respectively. By optimizing the *response-based distillation loss* to mimic the prediction behaviors of teacher, and the *feature relation-based distillation loss* to learn from the embeddings generated by the teachers, the discourse structures information can be distilled into the task of propaganda identification.

#### 3.2.1 Learning Layers

Three types of learning layers are built on top of sentence $s_i$ or token embedding $w_{ij}$: propaganda learning layer, student discourse relation learning layer, and student discourse role learning layer.

The *propaganda learning layer* is to learn the main task of propaganda identification at either

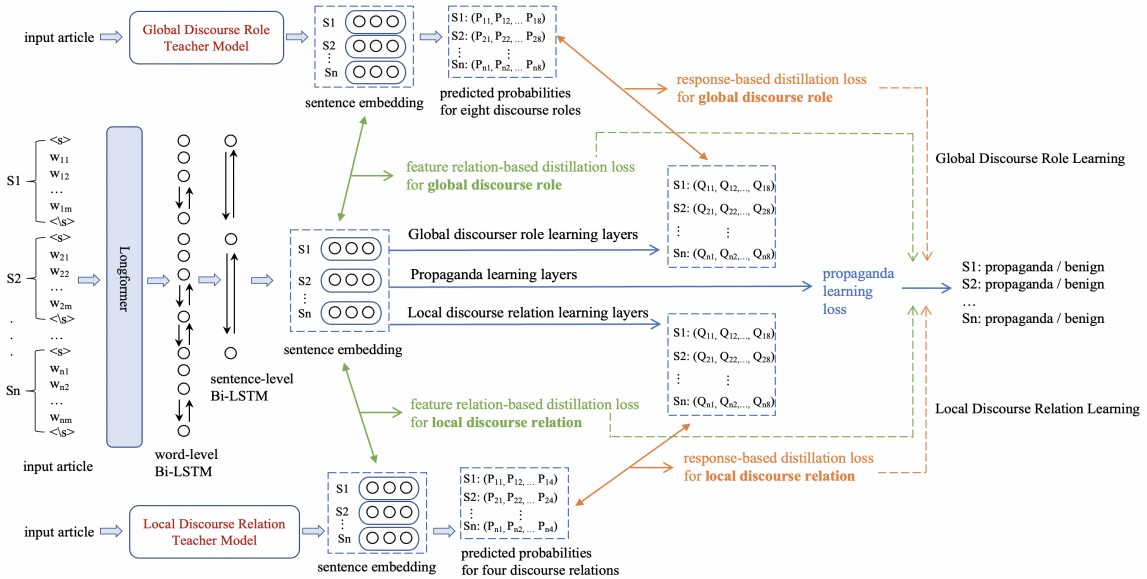

Figure 3: An illustration of propaganda identification guided by discourse structures via knowledge distillation

sentence level or token level:

$$Q_i^{propa} = softmax(W_2(W_1 s_i + b_1) + b_2)$$
$$Q_{ij}^{propa} = softmax(W_2(W_1 w_{ij} + b_1) + b_2) \quad (4)$$

where $Q_i^{propa}$ and $Q_{ij}^{propa}$ are the predicted probability of $i$-th sentence and its $j$-th token containing propaganda. $W_1, W_2, b_1, b_2$ are trainable parameters. The cross entropy loss is used for training:

$$Loss_{sent-propa} = -\sum_{i=1}^{n} P_i^{propa} \log(Q_i^{propa})$$

$$Loss_{token-propa} = -\sum_{i=1}^{n} \sum_{j=1}^{m} P_{ij}^{propa} \log(Q_{ij}^{propa}) \quad (5)$$

where $P_i^{propa}$ and $P_{ij}^{propa}$ are human annotated propaganda label for $i$-th sentence and its $j$-th token.

The *student discourse relation learning layer* is built on top of the concatenation of $i$-th sentence embedding $s_i$ and its adjacent sentence embedding $s_{i-1}$, to learn the discourse relation between them from the teacher model:

$$Q_i^{local} = (Q_{i1}^{local}, Q_{i2}^{local}, ..., Q_{i4}^{local})$$
$$= softmax(W_6(W_5(s_i \oplus s_{i-1}) + b_5) + b_6) \quad (6)$$

where $W_5, W_6, b_5, b_6$ are trainable parameters in the student discourse relation layer, $Q_i^{local}$ is the learned outcome of predicting discourse relations.

The *student discourse role learning layer* is built on top of the sentence embedding $s_i$, to learn its discourse role information from the teacher model:

$$Q_i^{global} = (Q_{i1}^{global}, Q_{i2}^{global}, ..., Q_{i8}^{global})$$
$$= softmax(W_4(W_3 s_i + b_3) + b_4) \quad (7)$$

where $W_3, W_4, b_3, b_4$ are trainable parameters in the student discourse role layer, and $Q_i^{global}$ is its learned outcome of predicting eight discourse roles.

### 3.2.2 Response-based Distillation

The response-based distillation loss (Hinton et al., 2015) is designed to minimize the discrepancy between the learned outcome of student layers and the predicted probability generated by the teacher models. By guiding the student layers to mimic the prediction behaviors of teachers, the knowledge of discourse relation and discourse role from the teachers can be distilled into the model.

Specifically, the Kullback–Leibler (KL) divergence loss is employed for measuring the distance between the learned probability of student layers and referenced probability from teacher models:

$$Loss_{response-local} = \sum_{i=1}^{n} P_i^{local} \log\left(\frac{P_i^{local}}{Q_i^{local}}\right) \quad (8)$$

$$Loss_{response-global} = \sum_{i=1}^{n} P_i^{global} \log\left(\frac{P_i^{global}}{Q_i^{global}}\right) \quad (9)$$

where $P_i^{local}$ and $P_i^{global}$ are response from the teachers, and are referenced as learning target. $Q_i^{local}$ and $Q_i^{global}$ are learned outcomes of student discourse relation layers and student discourse role layers. The response-based distillation loss penalizes the performance gap between teacher models and student layers, and forces student layers to be updated with discourse structures knowledge.

### 3.2.3 Feature Relation-based Distillation

The feature relation-based distillation loss is designed to seek guidance from the teacher-generated sentence embeddings which also contain discourse

|           | Comparison | Contingency | Temporal | Expansion | Macro |
|-----------|-----------|-------------|----------|-----------|-------|
| Precision | 85.75     | 80.06       | 86.42    | 82.17     | 83.60 |
| Recall    | 83.73     | 74.93       | 90.32    | 85.38     | 83.59 |
| F1-score  | 84.73     | 77.41       | 88.33    | 83.75     | 83.55 |

Table 3: Performance of the PDTB discourse relations model (local discourse relation teacher) on PDTB 2.0 dataset.

|           | M1    | M2    | C1    | C2    | D1    | D2    | D3    | D4    | Macro |
|-----------|-------|-------|-------|-------|-------|-------|-------|-------|-------|
| Precision | 55.56 | 37.88 | 43.72 | 67.21 | 66.67 | 62.69 | 75.22 | 62.15 | 63.23 |
| Recall    | 59.78 | 32.47 | 33.10 | 64.06 | 85.22 | 69.54 | 69.75 | 69.63 | 64.36 |
| F1-score  | 57.59 | 34.97 | 37.68 | 65.60 | 74.81 | 65.94 | 72.38 | 65.68 | 63.49 |

Table 4: Performance of the news discourse structure model (global discourse role teacher) on NewsDiscourse dataset. M1: Main Event, M2: Consequence, C1: Previous Context, C2: Current Context, D1: Historical Event, D2: Anecdotal Event, D3: Evaluation, D4: Expectation

structures knowledge. However, sentence embedding itself has no absolute meaning and instead relies on its spatial relations with other contexts. Thus, rather than directly minimizing the euclidean distance between teacher-generated and student-learned features, we follow (Park et al., 2019) to guide the student layers to learn the spatial relations between sentences found in the teacher models.

Specifically, let $s_i^{local}$ and $s_i^{global}$ denotes the $i$-th sentence embedding trained by the two teachers. The spatial matrix of the teachers are computed:

$$M_{ik}^{local} = cosine(s_i^{local}, s_k^{local})$$
$$M_{ik}^{global} = cosine(s_i^{global}, s_k^{global})$$
(10)

where $M_{ik}^{local}$ and $M_{ik}^{global}$ are spatial relation between $i$-th and $k$-th sentence in the teachers. Also, the spatial matrix of student-learned features is:

$$M_{ik} = cosine(s_i, s_k)$$
(11)

The feature relation-based distillation loss is the mean squared error (MSE) loss between spatial matrix of teacher models and student layers:

$$Loss_{relation-local} = \sum_{i,k}(M_{ik}^{local} - M_{ik})^2$$
$$Loss_{relation-global} = \sum_{i,k}(M_{ik}^{global} - M_{ik})^2$$
(12)

To summarize, the response-based distillation and feature relation-based distillation mutually complement each other, with the former informed by teacher-predicted probabilities and the latter guided by teacher-generated embeddings.

### 3.2.4 Learning Objective

The total distillation loss for local discourse relation and global discourse role are:

$$Loss_{local} = Loss_{response-local} + Loss_{relation-local}$$
$$Loss_{global} = Loss_{response-global} + Loss_{relation-global}$$
(13)

The overall learning objective for identifying propaganda at sentence and token level are:

$$Loss_{sent} = Loss_{sent-propa} + Loss_{global} + Loss_{local}$$
$$Loss_{token} = Loss_{token-propa} + Loss_{global} + Loss_{local}$$
(14)

## 4 Experiments

### 4.1 Dataset

Acquiring human-annotated labels at fine-grained levels is challenging and expensive, leading to a limited resource of available datasets. In our subsequent experiments, we utilized the propaganda dataset published by (Da San Martino et al., 2019b) that provides human-annotated labels for propaganda contents. We adhere to the same train / dev / test splitting in the released dataset. This propaganda dataset was also used in the NLP4IF-2019 challenge (Da San Martino et al., 2019a), which featured tasks involving sentence-level identification and token-level classification. In this paper, we specifically concentrate on propaganda identification at both the sentence and token levels.

### 4.2 Teacher Models

The teacher model for discourse relation is trained on PDTB 2.0 dataset (Prasad et al., 2008). Following its official suggestion, sections 2-21, sections 22 & 24 and section 23 are used for train / dev / test set respectively. Table 3 displays the classification performance for the four discourse relations. On the other hand, the teacher model for discourse role is trained on News Discourse dataset (Choubey et al., 2020). The performance of classifying the eight news discourse roles is presented in Table 4.

| | Sentence-level | | | Token-level | | |
|---|---|---|---|---|---|---|
| | Precision | Recall | F1 | Precision | Recall | F1 |
| Baseline Models | | | | | | |
| all-propaganda | 24.86 | 100.00 | 39.82 | 10.41 | 100.00 | 18.86 |
| chatgpt | 58.26 | 34.72 | 43.51 | 13.37 | 19.31 | 15.80 |
| chatgpt + 5-shot | 56.42 | 37.34 | 44.94 | 14.68 | 20.84 | 17.22 |
| chatgpt + discourse structures prompt | 57.93 | 38.61 | 46.34 | 15.82 | 21.96 | 18.39 |
| (Da San Martino et al., 2019b) | 63.20 | 53.16 | 57.74 | 39.57 | 36.42 | 37.90 |
| (Da San Martino et al., 2019a) | 60.28 | 66.48 | 63.23 | - | - | - |
| (Fadel et al., 2019) | - | - | 61.39 | - | - | - |
| (Vlad et al., 2019) | 59.95 | 57.47 | 58.68 | - | - | - |
| longformer | 60.32 | 60.50 | 60.41 | 34.60 | 39.81 | 37.03 |
| Feature Concatenation Models | | | | | | |
| + local discourse relation | 61.72 | 62.09 | 61.90 | 35.38 | 41.92 | 38.37 |
| + global discourse role | 61.50 | 63.58 | 62.52 | 36.39 | 41.28 | 38.68 |
| + both discourse structures | 62.71 | 64.08 | 63.38 | 36.62 | 42.27 | 39.25 |
| Knowledge Distillation Models | | | | | | |
| + local discourse relation | 60.40 | 66.17 | 63.15 | 35.18 | 43.49 | 38.90 |
| + global discourse role | 61.88 | 66.86 | 64.27 | 37.65 | 43.32 | 40.28 |
| + both discourse structures (full model) | **61.22** | **69.75** | **65.21** | **37.22** | **46.86** | **41.48** |

Table 5: Performance of sentence-level and token-level propaganda identification guided by discourse structures. Precision, Recall, and F1 of the propaganda class are shown. The model with the best performance is **bold**.

## 4.3 Baseline Models

We include the following baselines for comparison:

- **all-propaganda**: a naive baseline that predicts all sentences / tokens into *propaganda*

- **chatgpt**: an instruction prompt (A.1) is designed for the large language model ChatGPT to automatically generate predicted labels for sentence / tokens in the same test set

- **chatgpt + 5-shot**: we add five examples of propaganda sentences and five examples of non-propaganda sentences into the prompt

- **chatgpt + discourse structures prompt**: we add the local discourse relation and global discourse role of each sentence into the prompt

- (Da San Martino et al., 2019a): we present the best performance achieved by the rank one team in the NLP4IF-2019 challenge, where the model was also trained on extensive corpora including Wikipedia and BookCorpus

- (Da San Martino et al., 2019b): where both sentence and token level propaganda identification tasks are performed

- (Fadel et al., 2019): pretrained ensemble learning is employed for sentence-level task

- (Vlad et al., 2019): a capsule model architecture is designed for sentence-level task

- **longformer**: we build a baseline that follows the same framework and is equivalent to our developed model without discourse structures

## 4.4 Experimental Setting

The model takes the entire news article as input, and predicts the label for each sentence or token into *propaganda* or *benign*. The AdamW (Loshchilov and Hutter, 2019) is used as the optimizer. The maximum length of input is set to 4096. The number of training epochs is 6. The learning rate is adjusted by a linear scheduler. The weight decay is set to be 1e-2. Precision, Recall, and F1 of *propaganda* class is used as evaluation metric.

## 4.5 Experimental Results

Table 5 shows the performance of sentence-level and token-level propaganda identification.

Comparing feature concatenation models with the longformer baseline, we observe that integrating discourse relations or discourse roles as additional features brings consistent improvements for precision and recall, at both the sentence and token level tasks. This underscores that these two types of discourse structures can provide beneficial insights for identifying propaganda contents.

| sentence | discourse relation | prediction |
|---|---|---|
| This suggested, of course, that giving advice on how to prevent another 9/11, and all other Jihadist attacks against America, is now against Facebook's 'community standards'. | Causal | P (propaganda \| baseline) = 0.33
P (propaganda \| full model) = 0.86 |

| sentence | discourse role | prediction |
|---|---|---|
| ICE Deputy Director Thomas Homan said in a statement Tuesday, "The Oakland mayor's decision to publicize her suspicions about ICE operations increased risk for my officers and alerted criminal aliens – making clear this reckless decision was based on her political agenda" | Evaluation | P (propaganda \| baseline) = 0.42
P (propaganda \| full model) = 0.91 |

Figure 4: Examples of our method succeed in solving false negative error. Both the red sentences contain propaganda.

Comparing knowledge distillation models with the longformer baseline, it is evident that distilling the knowledge of discourse relations and discourse roles leads to a notable increase in recall by 9.25% and a significant enhancement in F1 score by 4.8%. Furthermore, in comparison to the previous best performance reported in (Da San Martino et al., 2019a), our knowledge distillation model exhibits superior performance in both precision and recall, ultimately achieving state-of-the-art results.

Comparing knowledge distillation models with feature concatenation models, we can see that distilling the knowledge from teacher models demonstrates stronger ability to incorporate two types of discourse structures, surpassing the approach of simply adding extra features.

Comparing our full model with the large language model ChatGPT, there still remains noticable performance gap, especially the recall. Also, the gap is even larger in terms of token-level task. Providing ChatGPT with extra examples or discourse structures information in the prompt can boost the performance a little bit, but it still remains inferior to our developed method.

### 4.6 Ablation Study

The ablation study of local discourse relation and global discourse role is also shown in Table 5. Both the two types of discourse structures play an essential role in identifying propaganda content, at both the sentence and token level tasks. Incorporating the two discourse structures together can further boost recall and achieves the best performance.

### 4.7 Effect of the Two Distillation Losses

Moreover, we examine the effect of two types of distillation losses in Table 6. Both response-based distillation and feature relation-based distillation yield substantial improvements. This demonstrates that learning from teacher-predicted probabilities and teacher-generated embeddings mutually complement each other, acquiring an enhanced guidance from discourse structures.

|  | Precision | Recall | F1 |
|---|---|---|---|
| longformer | 60.32 | 60.50 | 60.41 |
| + response-based | 61.64 | 67.96 | 64.65 |
| + relation-based | 60.40 | 66.77 | 63.42 |
| + both (full model) | **61.22** | **69.75** | **65.21** |

Table 6: Ablation study of the two types of distillation losses: response-based and feature relation-based. Take sentence-level propaganda identification as an example.

### 4.8 Effect of the Four Local Discourse Relations

In addition, we study the effect of the four local discourse relations in Table 7. The results indicate that removing any one of the four discourse relations leads to a performance drop compared to the full model, as expected, the influence of expansion relations is relatively less compared to the other three types of relations.

|  | Precision | Recall | F1 |
|---|---|---|---|
| longformer | 60.32 | 60.50 | 60.41 |
| the full model | **61.22** | **69.75** | **65.21** |
| - comparison | 60.91 | 67.76 | 64.15 |
| - contingency | 60.89 | 67.56 | 64.06 |
| - temporal | 61.03 | 68.16 | 64.38 |
| - expansion | 61.19 | 68.65 | 64.70 |

Table 7: Effect of removing each one of the four local discourse relations from the full model. Take sentence-level propaganda identification as an example.

### 4.9 Qualitative Analysis

Figure 4 presents examples of solving false negative error through the integration of discourse structures. The first propaganda sentence is inaccurately predicted as *benign* by the longformer baseline. However, by incorporating the local causal discourse relation into the model, the prediction is corrected to *propaganda*. Likewise, the second propaganda sentence is initially misclassified as a false negative by the baseline model. However, by leveraging the knowledge from the teacher model that this sentence plays an evaluation role in the article, the model successfully rectifies this error.

## 5 Related Work

**Propaganda** attracted research interests for years. Prior work focus on detecting propaganda at article-level (Rashkin et al., 2017; Barron-Cedeno et al., 2019). The first work on fine-grained propaganda analysis was introduced by (Da San Martino et al., 2019b,a). A shared challenge focusing on token-level tasks was launched by (Da San Martino et al., 2020). Several approaches have been developed for propaganda analysis, such as (Vlad et al., 2019) designed an unified neural network, (Fadel et al., 2019) utilized pretrained ensemble learning, (Dimitrov et al., 2021) trained a multimodal model mixing textual and visual features, and (Vijayaraghavan and Vosoughi, 2022) employed multi-view representations. In this paper, we focus on identifying propaganda in news articles at both sentence-level and token-level, leveraging discourse structures.

**Misinformation Detection** was also studied for years, such as fake news (Pérez-Rosas et al., 2018; Oshikawa et al., 2020), rumor (Wei et al., 2021; Li et al., 2019), political bias (Baly et al., 2020; Chen et al., 2020), and logical fallacy (Jin et al., 2022; Alhindi et al., 2022). Although propaganda may intersect with fake news, political bias, and logical fallacies, however, they are all distinct phenomena and tasks. Fake news and rumor always hallucinate untruthful information. Political bias refers to selectively reporting verified facts while leaving readers to arrive at their own conclusions. Logical fallacies focus on errors in reasoning and argumentations to reach an invalid conclusion. In contrast, propaganda presents unverified speculation or projections in the same tone as facts, and employs a variety of persuasion strategies to convince the readers, with the purpose to manipulate public beliefs to a predetermined conclusion.

**Media Bias**. In the most broad sense, propaganda news articles is a type of biased news reports. However, media bias often refers to ideological bias these days (Kiesel et al., 2019; Fan et al., 2019; Lei and Huang, 2022), and ideological bias is often expressed in a subtle way or under a neutral tone (van den Berg and Markert, 2020; Lei et al., 2022) by selectively including certain facts to subtly shift public opinions (Fan et al., 2019). In contrast, propaganda is not limited to hyper-partisan cases and can be applied to influence public beliefs in a way that aligns with the interests of the propagandist (Stanley, 2015; Rashkin et al., 2017). Propaganda often contains intensely emotional or opinionated content to incite or persuade the public (Da San Martino et al., 2019b), or presents unverified speculations, projections and deceptions (Miller and Robinson, 2019; Brennen, 2017). Indeed, in the current media landscape, ideologically biased media sources and propaganda media sources are often labeled separately, for example, Media Bias/Fact Check[2] distinguishes ideologically biased sources, conspiracy theory sources, questionable sources which includes major propaganda sources, and a couple other categories. Ideology bias and propaganda are studied separately as well in the NLP community (Barron-Cedeno et al., 2019; Liu et al., 2022), and each task features their own benchmark datasets (Fan et al., 2019; Baly et al., 2020; Da San Martino et al., 2019b) with documents retrieved from different media sources.

## 6 Conclusion

This paper aims to identify propaganda at sentence-level and token-level. We propose to incorporate two types of discourse structures into propaganda identification: local discourse relation and global discourse role. We further design a feature concatenation model and a knowledge distillation model to leverage the guidance from discourse structures.

## Limitations

This paper specifically concentrates on the identification of propaganda as a specific form of misinformation. There still exists various other forms of misinformation, such as fake news, conspiracy theories, and more. While the designed discourse structures method has demonstrated its usefulness in identifying propaganda, its effectiveness for other types of misinformation remains unknown.

## Ethics Statement

This paper focuses on the detection of propaganda, which falls within the broader category of misinformation and disinformation. The release of code and models should be utilized for the purpose of combating misinformation and not for spreading further misinformation.

## Acknowledgments

We would like to thank the anonymous reviewers for their valuable feedback and input. We gratefully acknowledge support from National Science

---

[2] https://mediabiasfactcheck.com/

Foundation via the awards IIS-1942918 and IIS-2127746.

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

# A  Appendix

## A.1  ChatGPT Prompt

The designed instruction prompt for sentence-level propaganda identification task is: "Propaganda is a form of misinformation or deceptive narratives that incite or mislead the public, usually with a political purpose. Please reply Yes if the following sentence contains propaganda content, else reply No. Sentence: "xxx". Answer:"

The designed instruction prompt for token-level propaganda identification task is: "Propaganda is a form of misinformation or deceptive narratives that incite or mislead the public, usually with a political purpose. Please extract the word in the following sentences that contains propaganda content. Please mimic the following output style. Example: "Of course, no "mistake" had occurred, the ban has been lifted only because of the wide publicity that we engaged in.". Words: wide, publicity. Sentence: "xxx". Words:"