# OpenReview forum: "Discourse Structures Guided Fine-grained Propaganda Identification"
_EMNLP/2023/Conference — EMNLP 2023 Main_

### Official Review · Reviewer_qWvx · 2023-08-02

**Typos Grammar Style And Presentation Improvements:** NA.
**Soundness:** 3

**Excitement:**

4: Strong: This paper deepens the understanding of some phenomenon or lowers the barriers to an existing research direction.

**Missing References:**

NA.

**Paper Topic And Main Contributions:**

This paper explores using local discourse relations and global discourse roles to improve propaganda detection.

**Questions For The Authors:**

NA.

**Reasons To Accept:**

- The paper makes a valuable contribution to the Linguistic side of NLP by uncovering the insight that certain discourse relations can lead to propaganda.
- The experiments are substantial, but there are implementation and theoretical issues that need to be addressed.

**Reasons To Reject:**

Weaknesses (W) of the paper:

W1: The claim for token-level analysis is weak. Despite the authors' assertion that token-level information is included, the paper lacks evidence pinpointing specific lexical choices or phrases that contribute to propaganda.

W2: I am highly skeptical about the reported performance of the discourse relation classification (Table 3). The authors claim over 80% accuracy for all 1st level sense classification in PDTB 2.0, whereas the state-of-the-art performance is at most 70%, with many relations performing below 60%. Notably, [1] and [2] specifically address this topic, making it unlikely for this paper, where discourse relation classification is only a sub-component, to outperform them. This raises concerns about potential implementation bugs or incorrect evaluation methods.

W3: There appears to be a lack of synergy between learning global and local discourse structure. The ablation analysis in Table 6 is superficial and fails to explain the reasons behind this synergy. The absence of even a simple example to shed light on this issue is concerning.

W4: The overall theoretical foundation of the paper requires further development. While the observation that Contrast / Contingency relations are often used for propaganda is noteworthy, it falls short of providing a holistic theory to support this claim. The paper attempts to link this observation with global discourse roles, but the analysis lacks evidence of the "cross feature" between local and global features that leads to propaganda. As a result, it seems that a discourse relation classifier might not be necessary, and instead, a contrast / contingency detector should be considered, as there is no discussion on expansion / temporal relations. The soundness of this aspect of the paper requires significant improvement, leading to a recommendation of a low soundness score for this manuscript, despite the idea is still exciting and worth encouragement.

References:

[1] A Side-by-side Comparision of Transformers for English Implicit Discourse Relation Classification  https://aclanthology.org/2023.codi-1.2.pdf

[2] Infusing Hierarchical Guidance into Prompt Tuning: A Parameter-Efficient Framework for Multi-level Implicit Discourse Relation Recognition  https://aclanthology.org/2023.acl-long.357.pdf

**Reproducibility:**

3: Could reproduce the results with some difficulty. The settings of parameters are underspecified or subjectively determined; the training/evaluation data are not widely available.

**Reviewer Confidence:**

4: Quite sure. I tried to check the important points carefully. It's unlikely, though conceivable, that I missed something that should affect my ratings.

---

> ### Author Rebuttal · Authors · 2023-08-29
>
> Thanks for your valuable questions! Here are our answers (A) to your questions (Q):
>
> **Q1:** I am highly skeptical about the reported performance of the discourse relation classification (Table 3). The authors claim over 80% accuracy for all 1st level sense classification in PDTB 2.0, whereas the state-of-the-art performance is at most 70%, with many relations performing below 60%. Notably, [1] and [2] specifically address this topic, making it unlikely for this paper, where discourse relation classification is only a sub-component, to outperform them. This raises concerns about potential implementation bugs or incorrect evaluation methods.
>
> **A1:** We are glad to clarify this misunderstanding.
> - PDTB 2.0 dataset distinguishes two types of discourse relations: explicit relations where the relation of two sentences are signified by explicit discourse connectives, and implicit relations where discourse connectives are omitted. At the coarse-grained level with the four discourse relations, the F1 score on explicit discourse relation classification can reach 90% or higher. Therefore, recent research on PDTB discourse relation classification primarily focuses on implicit relations. Both of the two papers your pointed to work on implicit relations as their titles indicate this as well. But, as stated in section 2.1.3 (line 178-181), **both explicit and implicit relations in the PDTB 2.0 dataset are used for training our discourse relation model.**
> - To further alleviate your concern, we report the performance of our discourse relation model on explicit and implicit relations separately in the table below:
>
>
> |  | Precision | Recall | F1-score |
> | -------- | -------- | -------- | -------- |
> | Explicit relations | 96.62 | 96.49 | 96.55 |
> | Implicit relations | 61.50 | 60.60 | 61.05 |
> | Both explicit and implicit (reported in the paper) | 83.60 | 83.59 | 83.55 |
>
>
> [1] A Side-by-side Comparision of Transformers for English **Implicit** Discourse Relation Classification https://aclanthology.org/2023.codi-1.2.pdf \
> [2] Infusing Hierarchical Guidance into Prompt Tuning: A Parameter-Efficient Framework for Multi-level **Implicit** Discourse Relation Recognition https://aclanthology.org/2023.acl-long.357.pdf
>
>
>
>
> **Q2:** There appears to be a lack of synergy between learning global and local discourse structure. The ablation analysis in Table 6 is superficial and fails to explain the reasons behind this synergy. The absence of even a simple example to shed light on this issue is concerning. The paper attempts to link this observation with global discourse roles, but the analysis lacks evidence of the "cross feature" between local and global features that leads to propaganda.
>
>
> **A2:** We address your question as follows:
> - **The ablation study in Table 5 shows that both local and global discourse structures are essential for identifying propaganda.** We did conduct the ablation study of local discourse relation and global discourse role in Table 5 and illustrate in section 4.6 (line 466-472). We actually finished two types of ablation study in the paper: the first is studying the effect of local and global discourse structures (Table 5); the second is studying the effect of two knowledge distillation strategies (Table 6).
> - **The statistical analysis in Table 1 and Table 2 proves that both local and global discourse structures provide auxiliary information for identifying propaganda.** Statistical analysis in Table 1 (section 2.1.2) and Table 2 (section 2.2.2) confirm that different local discourse relations and global discourse roles present different likelihoods of carrying propaganda, and thus can provide auxiliary information. These statistical analysis provide evidence for us to incorporate both local and global discourse structures into propaganda discovery.
> - **We follow your suggestion to add an analysis example to show the synergy effect of local and global discourse structures.** For the propaganda sentence S2 in the example article 1 of Figure 1: "*This suggested of course that giving advice on how to prevent another 9/11 is now against Facebook's community standards.*", the probability of correctly classifying as propaganda of baseline model, adding local discourse relation, add global discourse role, as well as adding them both are shown in the table below. Incorporating either local or global discourse structure can increase the probability confidence and solve false negative error, while integrating them both yields the best confidence.
>
>
>
> | Model | Discourse Structures Information | Prediction |
> | -------- | -------- | -------- |
> | baseline | None | P (propaganda \| baseline) = 0.33 |
> |+ local discourse relation | local discourse relation is *causal* | P (propaganda \| + local) = 0.77 |
> |+ global discourse role | global discourse role is *evaluation* | P (propaganda \| + global) = 0.64 |
> | + both (full model) | both local and global | P (propaganda \| full model) = 0.86 |
>
>
>
>
> **Q3:** The overall theoretical foundation of the paper requires further development. While the observation that Contrast / Contingency relations are often used for propaganda is noteworthy, it falls short of providing a holistic theory to support this claim...It seems that a discourse relation classifier might not be necessary, and instead, a contrast / contingency detector should be considered, as there is no discussion on expansion / temporal relations...
>
> **A3:** We clarify your concern as follows:
> - In our introduction, we highlighted using Contrast / Contingency discourse relations for propaganda detection as our data analysis shows strong positive correlations between these two relations and propaganda, consistent with some relevant discussions in the literature (*Horz and Korhan 2022, Petee and Alexis 2020, Jowett and Victoria 2018, Shabo 2008, Ecker et al. 2022*). But **we actually analyzed all the four types of discourse relations as shown in Table 1, and noted in section 2.1.2** (line 161-169) that in addition to contingency and comparison relations, temporal relations are clearly useful for propaganda detection as well which however are negatively correlated with propaganda: “Table 1 shows the ratio of propaganda sentences that have each of the four discourse relations with nearby sentences. The numerical analysis confirms our observation: sentences that exhibit contingency and comparison relations with adjacent sentences are more prone to containing propaganda, whereas sentences that narrate events in a chronological order significantly contain less propaganda.”
> - **To further alleviate your concern, we did an extra ablation study focusing on the four relations, which demonstrates the usefulness of all the four discourse relations.** The ablation is on the sentence-level propaganda identification task, the following table shows that removing each one of the four relations leads to a performance drop compared to the full model, as expected, the influence of expansion relations is relatively less compared to the other three relations. The results prove the necessity and effectiveness of all the four relations.
>
>
> |  | Precision | Recall | F1-score |
> | -------- | -------- | -------- | -------- |
> | longformer (baseline) | 60.32 | 60.50 | 60.41 |
> | knowledge distillation model (full model) | **61.22** | **69.75** | **65.21** |
> | *comparison* relation removed | 60.91 | 67.76 | 64.15 |
> | *contingency* relation removed | 60.89 | 67.56 | 64.06 |
> | *temporal* relation removed | 61.03 | 68.16 | 64.38 |
> | *expansion* relation removed | 61.19 | 68.65 | 64.70 |
>
>
>
> >[1] Horz, Carlo, and Korhan Kocak. "How to keep citizens disengaged: Propaganda and causal misperceptions." (2022).\
> >[2] Petee, Maia, and Alexis Palmer. "UNTLing at SemEval-2020 Task 11: Detection of Propaganda Techniques in English News Articles." Proceedings of the Fourteenth Workshop on Semantic Evaluation. 2020.\
> >[3] Jowett, Garth S., and Victoria O'donnell. Propaganda & persuasion. Sage publications, 2018.\
> >[4] Shabo, Magedah. Techniques of propaganda and persuasion. Prestwick House Inc, 2008.\
> >[5] Ecker, Ullrich KH, et al. "The psychological drivers of misinformation belief and its resistance to correction." Nature Reviews Psychology 1.1 (2022): 13-29.
>
>
>
>
>
> **Q4:** The claim for token-level analysis is weak. Despite the authors' assertion that token-level information is included, the paper lacks evidence pinpointing specific lexical choices or phrases that contribute to propaganda.
>
> **A4:** We are glad to provide some representative examples for token-level propaganda identification, where the propagandistic spans are shown in italic:
> - Example 1: “Vichy journalism,” a term which now fits so much of the mainstream media - *it collaborates in the same way that the Vichy government in France collaborated with the Nazis.*
> - Example 2: His opinion is: “Take it seriously, but with a large grain of salt.” *Which is just Allen’s more nuanced way of saying: “Don’t believe it.”*
> - Example 3: It describes the tsunami of vindictive personal abuse that has been heaped upon Julian from well-known journalists, many claiming liberal credentials. The Guardian, *which used to consider itself the most enlightened newspaper in the country*, has probably been the worst.
>
> Overall, propagandistic spans are usually baseless, speculative, opinionated content with persuasion techniques to convince or mislead the readers. We will add this analysis to the paper.

---

### Official Review · Reviewer_xm9p · 2023-08-03

**Soundness:** 3

**Excitement:**

3: Ambivalent: It has merits (e.g., it reports state-of-the-art results, the idea is nice), but there are key weaknesses (e.g., it describes incremental work), and it can significantly benefit from another round of revision. However, I won't object to accepting it if my co-reviewers champion it.

**Missing References:**

- Interpretable Propaganda Detection in News Articles <https://aclanthology.org/2021.ranlp-1.179/>

- A Survey on Computational Propaganda Detection <https://www.ijcai.org/proceedings/2020/672>

- Predicting Factuality of Reporting and Bias of News Media Sources <https://aclanthology.org/D18-1389/>

- Predicting Sentence-Level Factuality of News and Bias of Media Outlets <Proceedings of RANLP 2023>

- Sentence-level Media Bias Analysis Informed by Discourse Structures <https://aclanthology.org/2022.emnlp-main.682/>

**Paper Topic And Main Contributions:**

In this paper, the authors propose a study on the impact of discourse structure to improve propaganda identification in political news article. As an advance, they propose a fine-grained approach based on sentence-level identification and token-level identification.

**Questions For The Authors:**

- I suggest to the authors replace “embued” with “embodied” or “embed”.

- Please, could the authors provide a reference for the Propaganda definition? (e.x. Lines 028-030).

- As I said before, the authors need to show clearly the difference between media bias detection and propaganda identification tasks in news political articles. For example, all examples presented by authors may be classified as media bias according to the AllSides guideline, which describes 16 different types of media bias <see https://www.allsides.com/media-bias/how-to-spot-types-of-media-bias>.  Please, could the authors clearly present the distinction between both tasks?

**Reasons To Accept:**

- This paper is well-written and well-presented

- The discourse framework (PDTB) used in this study is reliable and widely used by NLP literature. Furthermore, taking into consideration the nature of this task (propaganda identification), discourse structure is definitely a relevant strategy.

- The experimental settings seem correct.

**Reasons To Reject:**

- My main concern related to this paper consists of the propaganda identification task definition. All examples presented by authors in Figures 1 and 2 consist of examples related to the Media Bias Identification task. For example, according to a reliable guideline proposed by AllSides <https://www.allsides.com/unbiased-balanced-news>, there are 16 different types of media bias that may be found in news articles. And all examples presented by authors are defined by this guideline as media bias. Hence, I am afraid that this kind of content could be classified as biased, therefore it seems not propaganda in news political articles.

- Corroborating my concerns described above, the authors did not provide a reliable conceptualization of the propaganda task, as well as a reference for the definition of propaganda. For example, in lines 028 and 030, there is not any reference for the given definition of propaganda. Therefore, the authors seem unsure about the main task of this paper.

- Finally, there is a wide range of state-of-the-art propaganda identification references missing in this paper.

**Reproducibility:**

2: Would be hard pressed to reproduce the results. The contribution depends on data that are simply not available outside the author's institution or consortium; not enough details are provided.

**Reviewer Confidence:**

5: Positive that my evaluation is correct. I read the paper very carefully and I am very familiar with related work.

**Typos Grammar Style And Presentation Improvements:**

- I suggest to the authors replace “embued” with “embodied” or “embed”.

- I encourage the authors to replace “intelligence models” with “expert models” in line 042.

---

> ### Author Rebuttal · Authors · 2023-08-29
>
> Thanks for your valuable reviews! Here are our answers (A) to your questions (Q):
>
> **Q1:** My main concern related to this paper consists of the propaganda identification task definition. The authors did not provide a reliable conceptualization of the propaganda task, as well as a reference for the definition of propaganda.
>
> **A1:** We have seen a pretty consistent definition of propaganda in prior works. Usually, propaganda is a type of misleading and deceptive narratives that are typically disseminated and used to influence and manipulate public opinion, attitudes, beliefs, or behaviors in a way that aligns with the interests of the propagandist or the organization disseminating the information.
>
> And we will add the following references *Lasswell 1927*, *Henderson 1943*, *Bernays 2005*, *Stanley 2015*, *Rashkin et al. 2017*, *Barrón-Cedeno et al. 2019* for defining propaganda.
>
>
>
> >[1] Lasswell, Harold D. "The theory of political propaganda." American Political Science Review 21.3 (1927): 627-631.\
> >[2] Henderson, Edgar H. "Toward a definition of propaganda." The Journal of Social Psychology 18.1 (1943): 71-87.\
> >[3] Bernays, Edward L. Propaganda. Ig publishing, 2005.\
> >[4] Stanley, Jason. How propaganda works. Princeton University Press, 2015.\
> >[5] Barrón-Cedeno, Alberto, et al. "Proppy: Organizing the news based on their propagandistic content." Information Processing & Management 56.5 (2019): 1849-1864.\
> >[6] Rashkin, Hannah, et al. "Truth of varying shades: Analyzing language in fake news and political fact-checking." Proceedings of the 2017 conference on empirical methods in natural language processing. 2017.
>
>
>
> **Q2:** The authors need to show clearly the difference between media bias detection and propaganda identification tasks in news political articles. Could the authors clearly present the distinction between both tasks?
>
>
> **A2:** Thank you for your insightful comment! Media bias often refers to ideological bias these days (*Kiesel et al 2019*). Propaganda is not limited to hyperpartisan cases and its language can be more emotionally charged and may describe unverified speculations or deceptions (*Stanley 2015*). Further, many common techniques of propaganda as described in (*Da San Martino et al. 2019*) are seldom seen in bias news, such as flag waving, causal oversimplification, black-and-white fallacy and whataboutism (*Fan et al. 2019*). In short, based on our analysis, we noted two main differences between propaganda and media bias, which have also been discussed in prior work:
> - **Emotional vs. Neutral tone: Propaganda is emotional and opinionated to persuade the public, while media bias tends to express in a neutral tone.** Propaganda intensely inserts emotional or opinionated content to incite or persuade the public (*Da San Martino et al 2019*), while media bias is more often expressed in a neutral and rational way (*van den Berg and Katja 2020*).
> - **Baseless vs. Factual: Propaganda usually employs unproved and baseless allegations, while media bias selectively reports verified facts.** Propaganda usually presents unverified speculations, projections, or deceptions (*Miller and Piers 2019*; *Brennen 2017*) and detecting propaganda requires the discovery of different persuasion techniques (*Da San Martino et al 2019*; *Gupta et al 2019*; *Da San Martino et al 2020*). Media bias refers more to selective bias or framing bias, that selectively includes or omits certain facts to sway readers’ stances (*Fan et al. 2019*).
>
> Indeed, in the current media landscape, **propaganda media sources and biased media sources are labeled separately by media outlet and studied separately in the NLP community**. For instance, the Media Bias/Fact Check website organized media sources to ideologically left/right leaning sources, conspiracy sources, questionable sources which includes major propaganda sources, and a couple other categories. Looking into the existing datasets the community has created, bias news articles and propaganda news articles were retrieved from different media sources (*Barrón-Cedeno et al. 2019*; *Liu et al. 2022*).
>
>
> >[1] Miller, David, and Piers Robinson. "Propaganda, politics and deception." The Palgrave handbook of deceptive communication (2019): 969-988.\
> >[2] Brennen, Bonnie. "Making sense of lies, deceptive propaganda, and fake news." Journal of Media Ethics 32.3 (2017): 179-181.\
> >[3] Lisa Fan, Marshall White, Eva Sharma, Ruisi Su, Prafulla Kumar Choubey, Ruihong Huang, and Lu Wang. 2019. In Plain Sight: Media Bias Through the Lens of Factual Reporting. In Proceedings of the 2019 Conference on Empirical Methods in Natural Language Processing and the 9th International Joint Conference on Natural Language Processing (EMNLP-IJCNLP), pages 6343–6349, Hong Kong, China. Association for Computational Linguistics.\
> >[4] Da San Martino, Giovanni, et al. "Fine-grained analysis of propaganda in news article." Proceedings of the 2019 conference on empirical methods in natural language processing and the 9th international joint conference on natural language processing (EMNLP-IJCNLP). 2019.\
> >[5] van den Berg, Esther, and Katja Markert. "Context in informational bias detection." Proceedings of the 28th International Conference on Computational Linguistics. 2020.\
> >[6] Kiesel, Johannes, et al. "Semeval-2019 task 4: Hyperpartisan news detection." Proceedings of the 13th International Workshop on Semantic Evaluation. 2019.\
> >[7] Stanley, Jason. How propaganda works. Princeton University Press, 2015.\
> >[8] Barrón-Cedeno, Alberto, et al. "Proppy: A system to unmask propaganda in online news." Proceedings of the AAAI Conference on Artificial Intelligence. Vol. 33. No. 01. 2019.\
> >[9] Giovanni Da San Martino, Alberto Barrón-Cedeño, and Preslav Nakov. 2019. Findings of the NLP4IF-2019 Shared Task on Fine-Grained Propaganda Detection. In Proceedings of the Second Workshop on Natural Language Processing for Internet Freedom: Censorship, Disinformation, and Propaganda, pages 162–170, Hong Kong, China. Association for Computational Linguistics.\
> >[10] Pankaj Gupta, Khushbu Saxena, Usama Yaseen, Thomas Runkler, and Hinrich Schütze. 2019. Neural Architectures for Fine-Grained Propaganda Detection in News. In Proceedings of the Second Workshop on Natural Language Processing for Internet Freedom: Censorship, Disinformation, and Propaganda, pages 92–97, Hong Kong, China. Association for Computational Linguistics.\
> >[11] Giovanni Da San Martino, Alberto Barrón-Cedeño, Henning Wachsmuth, Rostislav Petrov, and Preslav Nakov. 2020. SemEval-2020 Task 11: Detection of Propaganda Techniques in News Articles. In Proceedings of the Fourteenth Workshop on Semantic Evaluation, pages 1377–1414, Barcelona (online). International Committee for Computational Linguistics.\
> >[12] Yujian Liu, Xinliang Frederick Zhang, David Wegsman, Nicholas Beauchamp, and Lu Wang. 2022. POLITICS: Pretraining with Same-story Article Comparison for Ideology Prediction and Stance Detection. In Findings of the Association for Computational Linguistics: NAACL 2022, pages 1354–1374, Seattle, United States. Association for Computational Linguistics.
>
>
>
>
>
>
>
> **Q3:** Finally, there is a wide range of state-of-the-art propaganda identification references missing in this paper.
>
> **A3:** Thanks for your suggested papers! We have included the previous state-of-art model that achieved the best performance as one of the baselines (*Da San Martino et al 2019*). The model in the suggested papers actually are inferior to this previous state-of-art model baseline. We will follow your suggestion to add them in the paper.
>
>
> >[1] Giovanni Da San Martino, Alberto Barrón-Cedeño, and Preslav Nakov. 2019. Findings of the NLP4IF-2019 Shared Task on Fine-Grained Propaganda Detection. In Proceedings of the Second Workshop on Natural Language Processing for Internet Freedom: Censorship, Disinformation, and Propaganda, pages 162–170, Hong Kong, China. Association for Computational Linguistics.

---

### Official Review · Reviewer_sTTb · 2023-08-07

**Soundness:** 4

**Excitement:**

4: Strong: This paper deepens the understanding of some phenomenon or lowers the barriers to an existing research direction.

**Missing References:**

-

**Paper Topic And Main Contributions:**

Summary:

This paper addresses the task of propaganda detections. They posit that modeling local and global discourse relations would help in performing better on the classification task of propaganda detection. To test their hypothesis, they first perform statistical analysis on the dev set. Then, they design neural classifiers based on LongFormer to train the models for learning PDTB relations between sentences as well as discourse roles of the sentences according to a pre-defined ontology. Then, they design two models that leverage these predictions to improve on the task of sentence level and token level propaganda prediction. In the first model, they concatenate the probability outputs from the discourse relation models. In the second model, they force the propaganda detection model to mimic the outputs of the discourse relation models while learning propaganda detection in a multi-task setting. They compare the performance of this model to existing baselines and show that it performs better than the baselines on propaganda detection. Then, they perform ablation study to observe that the two types of models complement each other in performance.

Overall, the paper investigates their hypothesis in a robust manner and show that the empirical evidence supports their hypothesis. They also provide examples and statistical analysis to motivate their approach. The paper could be improved by including additional datasets (hard to find for this task) and using the discourse relations and few-shot in-context learning paradigm better (by adding the discourse relation prediction to prompts and evaluating whether it improves the performance in that paradigm as well). But, these are extraneous to their line of evaluation and hence are not necessary for the main message of the paper.

Contributions:
1) Interesting investigation of usefulness of modeling discourse relations for propaganda detections.
2) Statistical analysis and empirical evaluation to back their hypothesis.
3) Empirical results and ablation study that demonstrate the effectiveness of the proposed models and their assembly.

**Questions For The Authors:**

1) Could you elaborate more on the missing document level evaluation. Is it the norm for the task to evaluate at sentence level  and token level?

**Reasons To Accept:**

Strengths:
1) The paper does a good job of evaluating their hypothesis using both statistical and empirical evaluation. Ablation study further shows the complementary nature of the proposed approaches and hence makes the technical and conceptual contribution of the paper substantial for the task. As I haven’t worked directly on the task of propaganda detection, I am unable to confidently claim the novelty but it is novel to the best of my knowledge on this task.
2) Results support their hypothesis and the baselines cover most of the existing state of the art models.
3) The task is well motivated and very relevant to current social scenario. Examples given in the paper make it very easy to understand the main hypothesis and the approach clearly.
4) The paper is well-written and very easy to follow and understand.

**Reasons To Reject:**

Weaknesses:
1) All of the empirical evaluation is performed on one dataset. This makes it hard to judge the generalizability of the approach. But, it is understandable given the difficulty in annotating data for such a task.
2) The chat-gpt baseline is very rudimentary. Few-shot approach isn’t tested. Also, including the discourse relation information in the prompts (probably in a Chain-of-Thought style approach) might yield good results. This will only add to the paper’s evaluation. But, it is extraneous to their line of evaluation as presented.
3) In addition to sentence level and token level performance, it would have been interesting to see document level evaluation of propaganda as well. It seems like a natural setting for the task which is missing in their evaluation.

**Reproducibility:**

4: Could mostly reproduce the results, but there may be some variation because of sample variance or minor variations in their interpretation of the protocol or method.

**Reviewer Confidence:**

3: Pretty sure, but there's a chance I missed something. Although I have a good feel for this area in general, I did not carefully check the paper's details, e.g., the math, experimental design, or novelty.

**Typos Grammar Style And Presentation Improvements:**

-

---

> ### Author Rebuttal · Authors · 2023-08-29
>
> Thanks for your valuable comments and suggestions! Here are our answers (A) to your questions (Q):
>
> **Q1:** All of the empirical evaluation is performed on one dataset. This makes it hard to judge the generalizability of the approach. But, it is understandable given the difficulty in annotating data for such a task.
>
> **A1:** To the best of our knowledge, the propaganda dataset (*Da San Martino et al. 2019*) used in the paper is the only dataset that annotates propaganda at both sentence-level and token-level.
>
>
> >[1] Da San Martino, Giovanni, et al. "Fine-grained analysis of propaganda in news article." Proceedings of the 2019 conference on empirical methods in natural language processing and the 9th international joint conference on natural language processing (EMNLP-IJCNLP). 2019.
>
>
>
> **Q2:** The chatgpt baseline is very rudimentary. Few-shot approach isn’t tested . Also, including the discourse relation information in the prompts  (probably in a Chain-of-Thought style approach) might yield good results. This will only add to the paper’s evaluation. But, it is extraneous to their line of evaluation as presented.
>
> **A2:**
> - We follow your suggestion to add another two baselines, and report the results for sentence-level and token-level propaganda identification in the table below.
>     - *chatgpt 5-shot baseline:* we add five examples of propaganda sentences and five examples of non-propaganda sentences into the prompt.
>     - *chatgpt + discourse structures prompt:* we add the local discourse relation and global discourse role of each sentence into the prompt.
>
> - The experimental results demonstrate that **chatgpt with extra examples or discourse structures information in the prompt can perform better, but still inferior to our developed method.**
>
>
> |          | Sentence-level | Token-level |
> | -------- | -------------- | ----------- |
> | | Precision / Recall / F1 | Precision / Recall / F1 |
> | chatgpt  | 58.26 / 34.72 / 43.51 | 13.37 / 19.31 / 15.80 |
> | *chatgpt 5-shot* | 56.42 / 37.34 / 44.94 | 14.68 / 20.84 / 17.22 |
> | *chatgpt + discourse structures prompt* | 57.93 / 38.61 / 46.34 | 15.82 / 21.96 / 18.39 |
> | longformer (baseline) | 60.32 / 60.50 / 60.41 | 34.60 / 39.81 / 37.03 |
> | knowledge distillation model (our method) | **61.22** / **69.75** / **65.21** | **37.22** / **46.86** / **41.48** |
>
>
>
> **Q3:** In addition to sentence level and token level performance, it would have been interesting to see document level evaluation of propaganda as well. It seems like a natural setting for the task which is missing in their evaluation. Could you elaborate more on the missing document level evaluation. Is it the norm for the task to evaluate at sentence level and token level?
>
> **A3:**
> - Compared to sentence and token level task, **document-level propaganda identification is a relatively easy task.** Prior work detects propaganda at document-level, which is a relatively easy task. The work in *Barrón-Cedeno et al. 2019* shows that identifying propaganda document can achieve 97.21 F1 score on TSHP-17 dataset (*Rashkin et al. 2017*), and 82.89 F1 score on QProp dataset (*Barrón-Cedeno et al. 2019*). In addition, the dataset we used consists of primarily news articles from propagandistic media sources (82.5% of article), and was designed for fine-grained propaganda identification.
> - **The propaganda research in recent years has focused on fine-grained level.** The works in recent years focus more on research propaganda at fine-grained level, such as sentence-level (*Da San Martino et al. 2019; Vlad et al. 2019; Fadel et al. 2019*), or token-level (*Da San Martino et al. 2020*). Fine-grained propaganda identification can underscore the propagandistic content and thereby provide detailed interpretable explanations for propaganda document.
>
>
>
> >[1] Barrón-Cedeno, Alberto, et al. "Proppy: Organizing the news based on their propagandistic content." Information Processing & Management 56.5 (2019): 1849-1864.\
> >[2] Rashkin, Hannah, et al. "Truth of varying shades: Analyzing language in fake news and political fact-checking." Proceedings of the 2017 conference on empirical methods in natural language processing. 2017.\
> >[3] Barrón-Cedeno, Alberto, et al. "Proppy: A system to unmask propaganda in online news." Proceedings of the AAAI Conference on Artificial Intelligence. Vol. 33. No. 01. 2019.\
> >[4] Da San Martino, Giovanni, et al. "Fine-grained analysis of propaganda in news article." Proceedings of the 2019 conference on empirical methods in natural language processing and the 9th international joint conference on natural language processing (EMNLP-IJCNLP). 2019.\
> >[5] Giovanni Da San Martino, Alberto Barrón-Cedeño, Henning Wachsmuth, Rostislav Petrov, and Preslav Nakov. 2020. SemEval-2020 Task 11: Detection of Propaganda Techniques in News Articles. In Proceedings of the Fourteenth Workshop on Semantic Evaluation, pages 1377–1414, Barcelona (online). International Committee for Computational Linguistics.\
> >[6] Vlad, George-Alexandru, et al. "Sentence-level propaganda detection in news articles with transfer learning and BERT-BiLSTM-capsule model." Proceedings of the second workshop on natural language processing for internet freedom: censorship, disinformation, and propaganda. 2019.\
> >[7] Fadel, Ali, Ibraheem Tuffaha, and Mahmoud Al-Ayyoub. "Pretrained ensemble learning for fine-grained propaganda detection." Proceedings of the second workshop on natural language processing for internet freedom: censorship, disinformation, and propaganda. 2019.

---

### Meta-Review · Area_Chair_uvPH · 2023-09-15

**Recommendation:** 4

**Metareview:**

This is a well-presented paper which connects local (sentence, token-level) propaganda signals with global discourse structure, via both statistical analyses and empirical evidence via discourse aware propaganda prediction models.

The contribution on the conceptual (local vs global aspects of propaganda) and empirical (strong evaluation) levels are strong and are likely of broader interest to researchers in the field of propaganda prediction.

One remaining concern with the paper is its contextualization in theory and related work, by (1) incorporating a clearer definition of propaganda and distinguishing it from the phenomenon of "media bias" and (2) a more structured and exhaustive review of relevant related work as per the detailed suggestions by reviewer xm9p.

---

### Decision · Program_Chairs · 2023-10-07

**Decision:**

Accept-Main

**Comment:**

This is a well-presented paper which connects local (sentence, token-level) propaganda signals with global discourse structure, via both statistical analyses and empirical evidence via discourse aware propaganda prediction models.

The contribution on the conceptual (local vs global aspects of propaganda) and empirical (strong evaluation) levels are strong and are likely of broader interest to researchers in the field of propaganda prediction.

One remaining concern with the paper is its contextualization in theory and related work, by (1) incorporating a clearer definition of propaganda and distinguishing it from the phenomenon of "media bias" and (2) a more structured and exhaustive review of relevant related work as per the detailed suggestions by reviewer xm9p.